# Recruitment in Health Services Research—A Study on Facilitators and Barriers for the Recruitment of Community-Based Healthcare Providers

**DOI:** 10.3390/ijerph181910521

**Published:** 2021-10-07

**Authors:** Franziska Krebs, Laura Lorenz, Farah Nawabi, Isabel Lück, Anne-Madeleine Bau, Adrienne Alayli, Stephanie Stock

**Affiliations:** 1Faculty of Medicine and University Hospital Cologne, Institute of Health Economics and Clinical Epidemiology (IGKE), University of Cologne, 50935 Cologne, Germany; Farah.Nawabi@uk-koeln.de (F.N.); Adrienne.Alayli@uk-koeln.de (A.A.); stephanie.stock@uk-koeln.de (S.S.); 2Platform Nutrition and Physical Activity (peb), 10115 Berlin, Germany; I.Lueck@pebonline.de (I.L.); am.bau@pebonline.de (A.-M.B.)

**Keywords:** recruitment, community-based healthcare providers, health services research

## Abstract

In health services research, the recruitment of patients is oftentimes conducted by community-based healthcare providers. Therefore, the recruitment of these healthcare providers is a crucial prerequisite for successful patient recruitment. However, recruiting community-based healthcare providers poses a major challenge and little is known about its influencing factors. This qualitative study is conducted alongside a health services research intervention trial. The aim of the study is to investigate facilitators and barriers for the recruitment of community-based healthcare providers. A qualitative text analysis of documents and semi-structured interviews with recruiting staff is performed. An inductive–deductive category-based approach is used. Our findings identify intrinsic motivation and interest in the trial’s aims and goals as important facilitating factors in healthcare provider recruitment. Beyond that, extrinsic motivation generated through financial incentives or collegial obligation emerged as a conflicting strategy. While extrinsic motivation might aid in the initial enrollment of healthcare providers, it rarely resulted in active trial participation in the long run. Therefore, extrinsic motivational factors should be handled with care when recruiting healthcare providers for health services research intervention trials.

## 1. Introduction

Ambulatory care is one major research field in health services research. Community-based practices are an especially important setting for research studies. In trials in the outpatient setting, the recruitment of patients is frequently conducted by community-based healthcare providers such as general practitioners or specialists. The recruitment of these healthcare providers is, therefore, a crucial prerequisite that can determine the success of a trial in health services research right from the start. The recruitment of patients via community-based healthcare providers provides the advantage of a comparatively easy access to the targeted patient group for researchers. However, unlike hospital-based healthcare providers, community-based healthcare providers operate independently, are not bound by instructions from a clinic director and are often not familiar with conducting and recruiting for research studies [1,2]. Thus, the recruitment of healthcare providers often proves to be a major challenge. As a result, trials frequently fail to reach the required sample size. Furthermore, recruitment problems can lead to delays in the schedule, increased trial costs and less conclusive results due to the decrease in statistical power [3]. Suitable and effective recruitment strategies are, therefore, needed to reach and attract healthcare providers for participation in trials. Various potential barriers to healthcare provider recruitment are reported in the literature. These comprise anticipated time barriers (particularly related to increased paperwork and enrollment procedures), data privacy concerns, concerns with regard to recruiting one’s own patients and the perception that the healthcare providers would have little involvement in the design of the trial [4,5]. Peer-to-peer recruitment, the use of existing networks, involvement in trial design, relevance of the research topic, perceived benefit for patients and low additional effort are, thus, discussed as beneficial for the recruitment of healthcare providers [6,7,8,9]. The role of other strategies, such as the use of (financial) compensation, remains unclear [10,11,12,13]. Existing studies on the recruitment of healthcare providers are subject to several limitations. This is because their results are drawn from surveys regarding healthcare providers’ general attitudes towards research or hypothetical participation in trials [4,14,15]. These designs hold high risks of bias, as hypothetical participation decisions do not inevitably lead to actual trial participation [16]. In addition to this, studies on recruitment processes frequently focus on the recruitment and retention of patients in trials [16,17,18]. There is still a lack of information on how to master healthcare provider recruitment as a first step towards patient recruitment in health services research trials. The current state of research in the field of recruitment is summarized by Bower et al. (2009): “Recruiting for science is not underpinned by a science for recruitment” [19]. Various initiatives launched by stakeholder groups and researchers in the field of trial methodology have also called for methods to improve recruitment for research and develop strategies for a better integration of trials into routine care [20,21]. To fill this gap in the existing research, this article describes findings on the process of recruiting community-based healthcare providers during a health services research intervention trial. 

This study identifies facilitators and barriers to the recruitment of community-based healthcare providers using the GeMuKi trial (acronym for “Gemeinsam gesund: Vorsorge plus für Mutter und Kind”—Strengthening health promotion: enhanced check-up visits for mother and child) as an example. Based on experiences gained in the GeMuKi trial, factors for the successful recruitment of healthcare providers for planning and conducting future trials in community-based settings are discussed.

## 2. Materials and Methods

### 2.1. Setting

The GeMuKi trial was designed as a hybrid-effectiveness-implementation trial (type II) and, therefore, collected data on effectiveness and implementation simultaneously [22]. It aimed at incorporating a structured, low-threshold lifestyle counseling intervention into routine prenatal visits and infant check-ups. The trial was funded by the Innovation Fund of the German Federal Joint Committee (G-BA). Details on the GeMuKi trial can be found elsewhere [23]. In short, trained gynecologists, midwives and pediatricians in the intervention group conducted brief counseling sessions using elements of motivational interviewing (MI). Data collection was conducted via a digital data platform [24]. For organizational reasons, assignment to intervention and control group was conducted on regional level rather than individual level, resulting in five intervention and five control regions. Pregnant women (n = 1860) were recruited by participating gynecologists in the study regions before the 12th week of gestation [23]. Since care for pregnant women in Germany is primarily provided in the outpatient setting by community-based gynecologists, gynecologist practices provide an ideal location in which to reach pregnant women for research purposes. The recruitment of gynecologists who, after being enrolled themselves, then recruited pregnant women was, therefore, crucial for the success of the trial. In Germany, community-based physicians are self-employed [25] and can, therefore, independently decide which additional programs they offer to their patients and whether or not to participate in health services research studies. 

Study coordinators, who were based in the study regions, carried out the entire recruitment process of community-based health care providers in the GeMuKi trial. This included identifying contact details within the sample frame of community-based healthcare providers, enrollment of healthcare providers into the trial and ongoing close support afterwards. During this process, the study coordinators established personal contact to all healthcare providers within the sample frame to discuss trial participation. All study coordinators held a degree in the fields of nutrition or sports science.

The GeMuKi trial’s recruitment process is illustrated in Figure 1. Eligible healthcare providers were identified based on the Association of Statutory Health Insurance Physicians (ASHIP) database, supplemented by internet searches. The final sample frame consisted of 818 gynecologists (513 in intervention regions und 305 in control regions). At the beginning, all healthcare providers were invited to information events. In total, 30 gynecologists attended (17 in intervention regions and 13 in control regions). After a constructive exchange at these events, advertising campaigns were launched to promote the trial within the study regions. For example, presentations at physician’s quality circles and *Stammtisch* discussions (regular, informal meetings outside of work) were held and, in addition to this, the study coordinators distributed mass information media such as flyers. Other tools used to publicize the trial included press articles and newsletters. All gynecologists in the intervention regions (n = 513) were invited to participate in a trial preparation workshop, which was a prerequisite for the intervention group to participate in the trial and deliver the intervention. For those who did not provide feedback on trial participation, the study coordinators conducted cold calls via phone and personal practice visits. A total of 141 gynecologists and 104 associated physicians’ assistants attended the trial preparation workshop. Gynecologists in the control regions did not receive training, as they were solely required to collect data and did not conduct the intervention themselves. After the workshops, the study coordinators sent reminders to all participants. In intervention and control regions, they visited the practices to provide on-site instruction on the digital data platform and trial organization. In conclusion, 63 (12% of those eligible) gynecologists in the intervention group and 65 (21% of those eligible) in the control group were, subsequently, enrolled in the trial. Finally, 36 gynecologists in the intervention group (57% of those enrolled) and 37 in the control group (57% of enrolled) actively recruited patients for the trial. The participating gynecologists received an incentive of EUR 100 per patient in the intervention group and EUR 40 per patient in the control group. By the end of recruitment, 792 patients had been recruited in the intervention regions and 674 patients in the control regions. 

During the trial process, adjustments were performed to the recruitment plan: two additional trial regions (one intervention and one control) were added to enlarge the sample frame. The total timeframe for the healthcare provider recruitment was 18 months. 

### 2.2. Study Design 

This qualitative study was conducted alongside the GeMuKi trial using a sequential design. Figure 2 provides an overview of the iterative data collection and the analytical approach. The report and conduct of the study was based on the ‘Consolidated criteria for Reporting Qualitative research’ (COREQ) (Appendix A) [26]. All data collection and analyses were conducted by the two first authors, both of whom held a master’s degree in the field of health sciences and sociology, respectively, and were experienced qualitative researchers.As a first step, a documentary analysis of internal project documents was performed to establish an overview of the factors that influence the recruitment process. Internal project documents are documents prepared as part of project implementation for use by members of the project team (e.g., meeting minutes, records of phone calls, etc.). Based on this, semi-structured interviews with the study coordinators, who were part of the project team and in charge of recruiting community-based healthcare providers, were conducted and analyzed. In the third step, all factors for successful recruitment of healthcare providers were discussed.

### 2.3. Data Sources

All data used in the study were collected after the recruitment of healthcare providers was completed (data collection started on 30 June 2020). For the documentary analysis, all available records (n = 137) were collected, such as documents from trial staff meetings, discussions with occupational associations and healthcare providers, and written project correspondence (see Appendix A for an overview of included documents). The collected documents were reviewed and included or excluded for further analysis depending on whether they contained information relevant to the recruitment process [27]. Of the 137 documents collected, 99 were included in the final analysis. In the second step, semi-structured interviews were conducted with the study coordinators. The researchers and study coordinators knew each other from their cooperation in the host trial and had a friendly working atmosphere. The topics of the interview guides were based on the results of the documentary analysis. The interview guide (see Appendix A) included questions based on the experience of the study coordinators. The objective of the interviews was to assess the various recruitment strategies and to gather information on the reasons why healthcare providers decided to participate or decline to participate in the trial. The interviews (*n* = 6) were performed via telephone due to COVID-19 contact restrictions. All study coordinators who worked in the GeMuKi trial were invited and agreed to participate in the interviews. Since interviews were conducted with all involved study coordinators, assessment of data saturation was not possible. Before the interview, the researchers outlined the aims and goals of the study to the interviewees. Field notes were taken by the researchers to record researcher’s impressions as well as features of the interaction. The average interview duration was 39 min (min = 20 min, max = 65 min). All the interviewees gave their written consent for digital recording of the interviews, further data processing and publication of results in the form of anonymized quotes. The interviews were recorded and analyzed anonymously.

### 2.4. Data Analysis

First, all data sources (internal documents and interviews) were analyzed separately and integrated at the data interpretation stage. The internal documents selected as relevant to the research topic were evaluated by means of qualitative text analysis. The authors used thematic analysis as described by Kuckartz (2014), which is a category-based method for the systematic analysis of qualitative data [28]. The researchers opted for an inductive approach; consequently, the construction of the categories was based solely on the collected data [28]. The results of the documentary analysis were used to inform the development of the interview guide. The data from the semi-structured interviews were transcribed and analyzed using thematic analysis in the MAXQDA 18 software (VERBI Software, Berlin, Germany). At this analysis step, a combination of deductive and inductive category constructions was deployed [29]. The deductive categories reflected the results of the previous documentary analysis. Consensual coding, a technique in which the material is independently coded by two researchers and then consensualized in an iterative process, was used [28]. The complex category system was visualized and was collaboratively discussed among the research team to sort, interpret and prioritize the results.

## 3. Results

The results for identified factors that promoted the recruitment of community-based healthcare providers were presented first, followed by factors that inhibited successful recruitment. Table 1 displays the final and comprehensive system of thematic categories. The results section summarizes the aspects that were most relevant for planning and conducting further health services research. The interviews were conducted and analyzed in German. Two researchers translated the quotes independently.

### 3.1. Facilitators for the Recruitment of Community-Based Healthcare Providers

All the interviewees described the intrinsic motivation of healthcare providers as the most important factor for active participation in the trial. For example, study coordinators provided the following assessments:


*“For them, the focus is on perinatal programming, so they also know what responsibility the physician has […] during pregnancy to address this […] Yes, they have understood the importance of these topics and it is important for them, and that is the main motivation to participate in GeMuKi.” (study coordinator 1_paragraph 16)*



*“I think that it plays an important role that there is an intrinsic motivation to participate in something like this, that an interest in this topic is given, because/ and that one also, yes, simply has the motivation to do more about this in day-to-day life.” (study coordinator 5_paragraph 10)*


Intrinsic motivation, thus, included an interest in the trial topics and a perception of them as important and relevant to regular care. It indicates the physicians’ need to improve the care provided to their own patients and to contribute to the development of their profession. Additionally, intrinsic motivation involves a general openness and curiosity with regard to new learnings and being up to date. The respondents also addressed extrinsic motivational factors that led to participation in the trial. These included: financial compensation, continuing medical education credits, regional peer group dynamics, and professional–political mandates. However, the respondents claimed that these factors played only a secondary role in the decision on active participation. Although some statements indicated that the financial compensation should have been higher, there is an agreement that the financial aspect was not a decisive reason for whether a healthcare provider participated. 


*“No one would have taken part for the sake of money, in order to pimp their salary a bit. I do not see that at all.” (study coordinator 6_paragraph 8)*


Some of the reported facilitating factors for recruitment related to the general set-up of a routine healthcare practice. For example, recruitment was reported to be easier if healthcare providers were already addressing the lifestyle topics as part of their regular care prior to entering the trial. All the interviewees cited convincing healthcare providers to participate in the trial within a short time frame as their most difficult task during the recruitment process. For example, they mentioned the importance of highlighting different information in the intervention and control groups and adapting their communication strategy accordingly. The amount of information relayed was, thus, scaled down to a minimum for busy practices, while more detailed explanations on the trial were provided when there was more time. Overall, the study coordinators emphasized the importance of efficient and charming communication when it came to recruitment:


*“When I was out and about a few times for cold calls, at the beginning you’re still a bit shy and at some point you know what you have to say to somehow get the people. So I think there is a lot of intuition and also empathy, on whom you encounter there and whether it then just falls on deaf or on open ears.” (study coordinator 5_paragraph 44)*


Interviewees agreed that, in terms of promoting the trial among gynecologists at the very beginning, visits to quality circles and *Stammtisch* events were beneficial for recruitment.

### 3.2. Barriers to the Recruitment of Community-Based Healthcare Providers

The major inhibiting factor was a lack of time. This factor resulted from the general set-up of a routine healthcare practice. In many cases, the study coordinators reported that there was no time for additional tasks that went beyond standard care during a busy everyday care routine. In addition to this, many practices were working at the limit of their capacity, so additional time spent on individual patients due to trial tasks resulted in other patients not being cared for. The study coordinators, therefore, saw the additional workload caused by the trial as the most critical barrier to recruitment. During the recruitment activities, study coordinators reported on problems arising of trial-related processes and the additional workload for gynecologists—enrollment, documentation and counselling—which was described as not being manageable. In this context, the interviewees also experienced the financial compensation for trial effort to be too low to provide an inducement. Another factor reported in this category was the digital implementation of trial components (digital data platform), which in some cases led to a rejection of participation.

Additionally, the study coordinators described barriers to recruitment that arose from the relationship with the healthcare providers’ professional association: the interviewees expressed their impression, that the actual target group, community-based gynecologists, did not feel sufficiently involved in the planning of the trial. Community-based healthcare providers in the study regions were not involved during the planning phase, though members of the German Professional Association of Gynecologists (Berufsverband der Frauenärzte) were present at trial meetings.

The interviewees problematized organizational aspects within the team of study coordinators. Interviewees reported that it was often not possible to obtain clear approvals or rejections for trial participation from healthcare providers, even after repeated contact attempts. In these cases, there was a lack of clarity as to how many contact attempts should be performed before a practice could be classified as not recruitable.


*“So I couldn’t tell the physician assistant anything more about it, she had already heard from me several times, HAD already presented everything to the physician […], but there was no final feedback. Then [it] was just: Okay, do I remove them from the list? Better not do it? That was always the decision. I think many of the study coordinators then immediately deleted the practice.” (study coordinator 1_paragraph 51)*


Another main difficulty in the recruitment work was seen in information management on the part of the physicians’ assistants. This included passing the information to the right person at the practice. In most cases, the initial telephone contact was conducted with physicians’ assistants. Often, the physician’s assistant acted as a gatekeeper. As a result of this, it was not possible to speak directly with the physician or practice owner. Frequently, the extent to which the information was passed on by the physician’s assistant was unclear.


*“[…] then you just have some physician’s assistant on the line. Well, they don’t tell you their NAME on the phone, they simply say “Practice such-and-such” and until you somehow get through to the one who is responsible […] That really sucks (laughs lightly) […]? If you then called them, they didn’t know about anything and until/ I was (…) VERY, VERY rarely put through to the physician at recruitment and […]/ I don’t even suggest that anymore. There’s no point.”[ (study coordinator 4_paragraph 10)*


### 3.3. Inactive Practices

Inactive practices are practices that enrolled in the trial but did not recruit patients. In the GeMuKi trial, this applied to 43% of all the enrolled practices (see Figure 1).

The interviewees reported a lack of intrinsic motivation and, in contrast, predominantly extrinsic motivational factors for initial trial enrollment, such as collegial obligations or continuing education credits for practices that were inactive from the very beginning: 


*“With the practices that (laughs lightly) only participate out of somehow a sense of duty, because they are regional leaders or something, because they have the feeling “Yes, okay, I have to enroll in a trial”, yes, or, yes, "I’m doing this here because it HAS to be somehow for the research", but who don’t have such a real passion behind it, with them it’s going slowly.” (study coordinator 6_ paragraph 34)*


Study coordinators mentioned that the reasons for practices becoming inactive during the trial were repeated rejection from patients and the perceived complexity of the trial, which led to implementation problems. According to the interviewees, rejection by patients was in some cases caused by health care provider’s lack of requisite arguments and techniques to convince eligible patients to participate in the trial. Furthermore, they reported that participating active healthcare providers felt abandoned in their region and become inactive due to frustration regarding the lack of engagement on the part of their colleagues.

## 4. Discussion

The aim of this article was to identify facilitators and barriers for the recruitment of community-based healthcare providers and to assess the recruitment strategies deployed in the GeMuKi trial.

Intrinsic motivation among healthcare providers clearly emerged as the most important prerequisite for actively participating in the trial. The importance of promoting intrinsic motivation has, likewise, been highlighted in previous studies on the recruitment of healthcare providers into trials [10,30,31]. When it comes to fostering intrinsic motivation, a strong emphasis should, thus, be placed on the added value of the trial [32]. Moreover, conducting an in-depth needs assessment within the target group of healthcare providers before conceptualizing a trial can be helpful in determining the fields of interest and perceived needs for the optimization of care [6]. This means that developing trial themes “bottom-up” can be used as a measure to increase the intrinsic motivation for trial participation among community-based healthcare providers [1,31,33].

In contrast, extrinsic motivating factors, such as financial incentives and collegial obligations, were shown to be overrated. The results of our study on financial compensation were inconsistent. While some healthcare providers called for higher financial compensation, study coordinators reported that financial compensation was not a motivator for active participation. In connection with this, no evidence of positive effects of peer-to-peer recruitment on recruitment rates was found in this study. This result was in contrast to previous research findings, highlighting the importance of peer-to-peer recruitment [9,13,34]. While in our study, this strategy did lead to trial enrollment in some cases, it rarely resulted in active trial participation in the long run. The high number of inactive practices tied up many resources, as multiple attempts were performed by the study coordinators to motivate these healthcare providers to recruit patients for the trial. It follows, that providers who lack intrinsic motivation should be ruled out at an early stage. 

In conclusion, extrinsic motivating factors emerged as a conflicting strategy when recruiting community-based healthcare-providers for an intervention trial. This result was unexpected, as extrinsic motivators such as peer-to-peer recruitment have been identified as beneficial in the literature. As the role of financial incentives remains unclear, more research is needed to assess the impact of this strategy on recruitment. The resulting issue of inactive practices that was found in this study might be unique to trials which place a high burden on participating healthcare providers. This is oftentimes the case in health services research when the intervention is carried out by healthcare providers themselves. In combination with a lack of intrinsic motivation, extrinsic motivating factors may create just enough engagement to enroll in the trial, but not enough to actively participate. However, published research investigating recruitment processes were mostly conducted within the frame of low-burden interventions. In this context the effects of extrinsic motivating factors can be completely different, leading to more beneficial effects of these strategies. When recruiting community-based healthcare providers for high-burden intervention trials, extrinsic motivating factors should be handled with care to avoid inactive practices in the enrolled sample.

Despite the results on the use of financial incentives for the active participation of health-care providers, financial incentives could still be regarded as a valuable tool in the process of recruiting physicians’ assistants for the trial. The physician’s assistant is generally the primary contact person for study personnel in the recruitment process; therefore, their cooperation and commitment is crucial. Information management on the part of the physicians’ assistants was identified as a barrier in this study and has also been reported previously by others [34,35,36]. The effectiveness of financial incentives to manage gatekeeping behavior should, therefore, be further researched. 

In addition to this, the barriers reported by healthcare providers should not be overestimated. Reported barriers may often be excuses for not participating or not recruiting patients into the trial [35,36,37]. Multiple adjustments after the start of the recruitment phase of the GeMuKi trial to address and overcome reported barriers cost many resources and, in the end, did not result in active participation on the part of healthcare providers. Hence, there seemed to be greater value in enhancing healthcare provider input during the planning phase of the trial and the recruitment strategy. By doing this, researchers could avoid barriers, create a sense of ownership and thereby build healthcare provider buy-in right from the start of the trial [1,6,30,32,38].

The findings of the study also emphasized the role of trial-related processes in healthcare providers’ recruitment decisions. Trial protocols that require a substantial change in the general setup of healthcare practice and/or involve complex tasks pose too great a hurdle for most healthcare providers, leaving only the most motivated for recruitment into the trial. When developing a trial, trialists should, therefore, aim for the smallest possible additional burden and level of change to current practice with which it is still possible to achieve the trial’s goals [13,32].

In the context of recruitment organization, the communication skills of the recruiting trial personnel were found to play a big role in recruitment. Effective and goal-oriented communication in recruitment was especially important during busy practice hours in community-based practice settings. As such, trial information must be adapted to different situations and actors, considering age, gender and professional status. Shortly after the start of recruitment, recruiting staff should reconsider which strategies have worked best and readjust as necessary. Effective communication between study sites and trial teams has been found to facilitate recruitment in other studies [6,30]. McDonald et al. proposed utilizing a business model approach and marketing techniques to foster trial recruitment [32]. This includes methods such as building brand values and adopting a formal marketing plan. To implement this approach, trial teams should prioritize these tasks and obtain expertise in the field of marketing and communication. 

Considering the issue of inactive practices (i.e., practices that were enrolled but did not actively participate by recruiting patients), a lack of recruitment skills of healthcare providers emerged as one key factor. In our study, healthcare providers did not recruit patients because they did not know how to introduce the trial and participation to their patients. Patient recruitment has previously been described as a ‘sales pitch’ [35,39], which poses a major challenge to healthcare providers. Furthermore, research shows that healthcare providers do not feel comfortable communicating the aims and design of the trial, do not want their patients to feel pressured to participate, and do not feel comfortable dealing with rejection [35,39,40]. Offering recruitment skills training in trial preparation workshops can overcome these barriers. The effectiveness of this strategy should, hence, be investigated further. Another strategy to counteract patient rejection, which can lead to frustration and the cease of patient recruitment on the side of the healthcare provider, is the use of comparatively high incentives for patients at the beginning of the trial. Options such as offering additional medical services are also conceivable as a viable incentive.

Community-based healthcare providers in Germany still only undertake trials rarely and lack research routines. To establish research structures in this setting, developing a network of research practices could be beneficial. The use of existing network structures for the recruitment of community-based physicians into trials has proven to be successful in other studies. In their quality of primary care trial, Wetzel et al. found general practitioner recruitment rates of 66% when recruiting from an established network, compared to 23% when these structures were not present [37]. It should be noted that recruiting from existing networks may induce sample effects and, therefore, lead to limitations in the generalizability of trial results [10,13,37]. However, the same argument also applies to a sample of healthcare providers who proactively engage in trials. These physicians are presumably more motivated to change current practice and do not represent the average physician in the field. Today, research practice networks are still rare in Germany and, if present, are limited to certain fields of expertise (e.g., family medicine). In the long term, aspects of conducting research and trial recruitment within routine care ought to be incorporated into the curriculum of community-based healthcare providers.

During the planning phase of the recruitment strategy in the GeMuKi trial, it became clear that advice on how to successfully recruit community-based healthcare providers was difficult to find. There was no doubt that parameters such as the trial design, the setting and the broader environment influenced the applicability and effectiveness of recruitment strategies. There are hardly any studies with a comparable research focus (prevention), in comparable settings (community-based physicians) and with a comparable trial burden on healthcare providers (recruiting patients, implementing, and performing an intervention, and documenting trial data). To better inform future health services research trials in recruitment planning, research should focus more on how the effectiveness of different recruitment strategies is influenced by these parameters.

### Strengths and Limitations

The presented findings were drawn from a large pragmatic controlled healthcare intervention trial and, therefore, represent recruitment issues under real-world conditions, which was an important strength of the study.

Another strength of this study was the combination of different methods and data sources. With this approach, it was possible to gain a comprehensive understanding and, thus, map the complexity of the recruitment process in the most accurate way.

One limitation was that information on recruitment was available only from healthcare providers who were accessible after the invitation to participate in the trial. Therefore, the barriers experienced by healthcare providers with whom it was not possible to establish contact after the initial invitation to the trial remain unknown. Moreover, the results of this study were based on the appraisals of six study coordinators and were, therefore, subjective in nature. It was not possible for the research team to gain direct access to healthcare providers to assess factors that influenced recruitment. As the recruiting trial staff was in contact with community-based healthcare providers on a daily basis, their experiences and perceptions were a valuable information source. The study described in this article was designed as a Study within a Trial (SWAT) [16]. As such, it was not possible to compare the effect of isolated recruitment strategies, as doing so would affect the scientific integrity of the host trial.

## 5. Conclusions

During the planning of a trial, more attention should be paid to the recruitment phase. Researchers should seek input from healthcare providers during the planning of the trial design and the recruitment strategy. It is advisable to conduct a thorough needs assessment to avoid barriers, address intrinsic motivation, and create a sense of ownership. Financial compensation for the trial burden emerged as a basic requirement, though this was not sufficient as a sole means of recruitment. Additionally, extrinsic motivational factors generally come with a risk of inactive participation. Moreover, clear, and goal-oriented communication skills on the part of trial staff were shown to positively influence recruitment. Sufficient preparation on how to introduce the trial to their patients is important for healthcare providers to feel adequately prepared for recruitment tasks. The recruitment skills of healthcare providers and the communication skills of the trial staff should, therefore, be addressed explicitly prior to the start of the recruitment phase.

## Figures and Tables

**Figure 1 ijerph-18-10521-f001:**
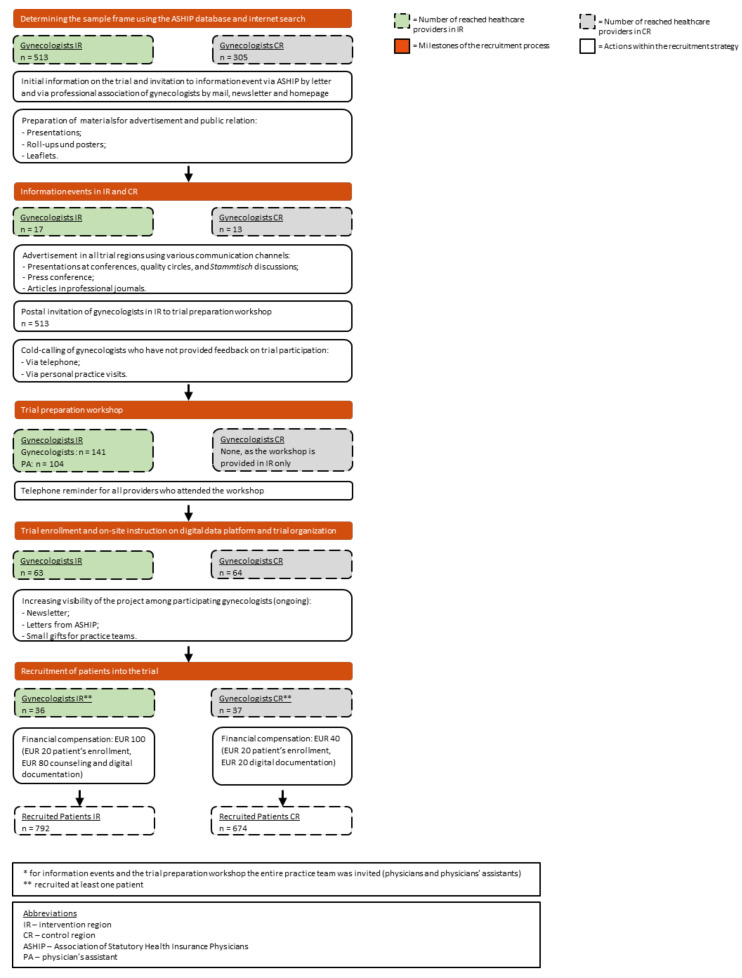
Flowchart for the recruitment process in the GeMuKi trial.

**Figure 2 ijerph-18-10521-f002:**
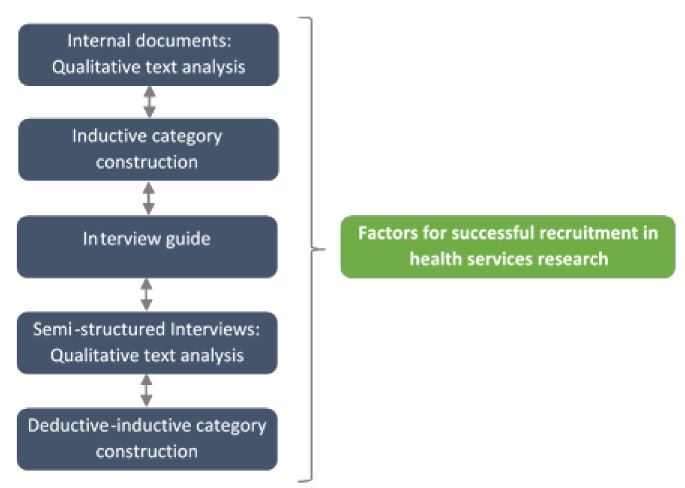
Iterative data collection and analytical approach.

**Table 1 ijerph-18-10521-t001:** Category system for thematic analysis.

Facilitators for the recruitment of community-based healthcare providers	Motivation for participation of healthcare providers	Intrinsic motivation	Relevance of the trial topic
			Professional development; improving care; support research
			Openness to learn something new/be up to date
			Improving professional cooperation
		Extrinsic motivation	Collegial obligation (generated by peer-to-peer recruitment)
			Committed to professional politics; professional–political mandates
			Financial Compensation
			Continuing education credits for informational event and training
	General set up of routine healthcare practice	Lifestyle topics were already part of regular care before entering the trial	
		Awareness that there is pent-up demand in medical care	
	Promising contact channels	Presentations at quality circles and Stammtisch events	
		Letters sent by the Association of Statutory Health Insurance Physicians (ASHIP)	
		Cold calls	
		Repeated personal visits combined with small presents for practice staff	
	Practice organization/distribution of tasks within the practice team	Coordination and communication within the practice teams	
		Participation of the physician’s assistant in trial tasks and close exchange with the gynecologist	
	Other facilitators	Individual characteristics of the healthcare providers	
		Efficient and charming communication and adapting communication to individual situation in the practice	
		Particularly high need among patients (practices in deprived areas)	
		Low trial burden	
Barriers for the recruitment of community-based healthcare providers	General set-up of routine healthcare practice	Lack of time and excessive workload in day-to-day routine	
		Lifestyle topics were NOT part of regular care before entering the trial	
		Information management on the part of the physicians’ assistants	
		Practice organization	Healthcare providers are reluctant to upset well-established practice structures
			Physicians’ assistants often work part-time. Trial tasks must, therefore, be carried out by several people
			Change of staff in the practice
			Rejection of the entire practice team
	Trial-related processes (inclusion and implementation)	Financial compensation is perceived as too low by some healthcare providers	
		Incentive for patients is perceived unattractive	
		Structure and content of the trail preparation workshop should be improved	
		Inclusion criteria sometimes not feasible in day-to-day practice	
		Digital data documentation: some practices only work paper-based	
	Professional policy	Target group in trial regions not included in planning (only professional associations)	
		Lack of support from the professional association	
	Organizational aspects within the team of study coordinators	Using the most appropriate communication and marketing strategies was difficult at the beginning	
		Uncertainty about frequency of repetitive cold calls and reminders	
	Participant clientele	Healthcare providers do not perceive any need for intervention among their well-educated patient clientele	
		Healthcare providers perceive that their socially vulnerable patient clientele has too many other burdens and cannot be reached by the intervention	
	Participant rejection	Healthcare providers have difficulties to “sell” the trial	
		Administrative effort too high and benefits too low	
		Characteristics of patients: both groups with high and low intervention needs	
		Data privacy concerns	
		No interest	
		Lack of trust between patient and healthcare provider	
		Recruitment at an unsuitable time point: uncertainty in early pregnancy leads to rejection	
	Other barriers	Individual characteristics of healthcare providers	
		Healthcare provider does not have any experience in recruiting patients	
		Adjustments to trial workflows were delayed by long bureaucratic processes	
		Skepticism regarding trials in general	
Explanations for inactive practices	No active participation at all	Enrollment out of obligation; no honest interest	
		Participation for receiving a free workshop and continuing education credits	
	Active participation discontinued during the trial	Frustration as colleagues in the region do not participate	
		Perceived complexity of the trial leads to problems and, ultimately, to healthcare providers quitting	
		Repeated rejection by patients to participate in the trial	
Unrelated discussion points and other matters	Suggestions for improvements		
	Expertise and knowledge exchange		

## Data Availability

The datasets used and analyzed in this study are available from the corresponding author on reasonable request.

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
