# Peer review of "Recruitment in Health Services Research—A Study on Facilitators and Barriers for the Recruitment of Community-Based Healthcare Providers"

_ijerph, 2021, doi:10.3390/ijerph181910521_

Round 1

Reviewer 1 Report

Overall, a very interesting, valuable paper that will inform future research practice globally in health service improvement. The paper is very well written and adequately addresses an issue in the community-based practice research sphere. The introduction provides a good background and overview of the current situation and states the aims of the study clearly. The methods a re well described and very well suited to the aims of the study. It is great to see good detail in the recruitment section. The results align nicely with the aims, although it might be a good idea to collate the major themes into a table - just as a summary. The discussion is high level, with good supporting literature and links to real world practice. The conclusions are aligned nicely with the aims and it is great to see both the strengths and limitations of the study highlighted. Well done. Some minor points to be corrected below. 

Line 87: Reference missing

Line 121: Some misalignment of text around table

Line 126: Reference missing

Line 145: Reference missing

Line 156: Reference missing

Line 161: Reference missing

Line 190: Reference missing

Line 282: Reference missing

Reviewer 2 Report

Even if the topic is increasingly relevant, the presentation of the results unfortunately at this stage is insuffient for publication. It needs

  1. in the methods section a clear structure and more stringent presentation
  2. a more detailed category-system and a much shorter presentation of major results (the option of supplementary material should be used)
  3. a short summary of major results at the beginning of the discussion and the clarification of what the study adds to existing knowledge.

Reviewer 3 Report

The draft has some potential, but I would not recommend the manuscript for publication in its present state. First of all, the manuscript is written poorly, so it is often difficult to understand what the authors want to convey. The manuscript should absolutely have been proofread before submission to the journal. Second, the biggest weakness of the draft is the depth of the research findings. Conducting clinical trials and health services research may include a wide range of issues including ethics, communications with patients, and so on. Research processes and methods are unconvincing to me studying barriers and facilitators for engaging community-based healthcare providers in health services research, so it is hard to buy the qualitative results. Finally, the literature review is insufficient so that readers may not be interested in this topic. Thus, the current manuscript needs to be further developed theoretically and empirically before being considered for publication.

Round 2

Reviewer 3 Report

The revised version appears to be much improved because the revision has been thoroughly made. I agree to accept this manuscript for publication.